# Clinical effectiveness and cost effectiveness of individual mental health workers colocated within primary care practices: a systematic literature review

Jean-Baptiste Woods [1], Geva Greenfield [2], Azeem Majeed [3], Benedict Hayhoe [2]

► Prepublication history and additional materials for this paper is available online. To view these files, please visit the journal online (http://dx.doi.org/10.1136/bmjopen-2020-042052).

¹Department of Primary Care and Public Health, School of Public Health, Imperial College London, London, UK
²Department of Primary Care & Public Health, Imperial College London, London, UK
³Primary Care, Imperial College London, London, UK

**Correspondence to**
Dr Geva Greenfield;
g.greenfield@imperial.ac.uk

## ABSTRACT

**Objectives** Mental health disorders contribute significantly to the global burden of disease and lead to extensive strain on health systems. The integration of mental health workers into primary care has been proposed as one possible solution, but evidence of clinical and cost effectiveness of this approach is unclear. We reviewed the clinical and cost effectiveness of mental health workers colocated within primary care practices.

**Design** Systematic literature review.

**Data sources** We searched the Medline, Embase, PsycINFO, Healthcare Management Information Consortium (HMIC) and Global Health databases.

**Eligibility criteria** All quantitative studies published before July 2019 were eligible for the review; participants of any age and gender were included. Studies did not need to report a certain outcome measure or comparator in order to be eligible.

**Data extraction and synthesis** Data were extracted using a standardised table; however, pooled analysis proved unfeasible. Studies were assessed for risk of bias using the Risk Of Bias In Non-randomised Studies - of Interventions (ROBINS-I) tool and the Cochrane collaboration's tool for assessing risk of bias in randomised trials.

**Results** Fifteen studies from four countries were included. Mental health worker integration was associated with mental health benefits to varied populations, including minority groups and those with comorbid chronic diseases. Furthermore, the interventions were correlated with high patient satisfaction and increases in specialist mental health referrals among minority populations. However, there was insufficient evidence to suggest clinical outcomes were significantly different from usual general practitioner care.

**Conclusions** While there appear to be some benefits associated with mental health worker integration in primary care practices, we found insufficient evidence to conclude that an onsite primary care mental health worker is significantly more clinically or cost effective when compared with usual general practitioner care. There should therefore be an increased emphasis on generating new evidence from clinical trials to better understand

## Strengths and limitations of this study

- ► This review's inclusion of a broad range of population types enables it to be more representative of the general practice setting the review is focused on.
- ► This review's inclusion of multiple different types of mental health worker can reduce the potential for the type of mental health worker being a confounding factor in whether significant outcome changes were identified.
- ► Non-English language studies were not included in this review; therefore, relevant evidence from non-English studies may have been missed.
- ► This study could have included a broader literature search that removed the database search terms relating specifically to integration, collocation and mental illnesses.
- ► Pooled analysis within this review proved unfeasible, due to the relatively small number of studies identified and the heterogeneity of effects and outcomes investigated.

the benefits and effectiveness of mental health workers colocated within primary care practices.

## INTRODUCTION

Mental health disorders contribute significantly to global disease incidence and prevalence, with one billion people affected by mental disorders[1] and 122.8 million disability-adjusted life years being lost annually through mental health problems.[2] The WHO argues that primary care is the optimal environment to treat patients with these disorders,[3] but research has shown that primary care practitioners may be inadequately treating a substantial proportion of these.[4 5] There have been many different quality improvement solutions envisioned in order to optimise the clinical and cost effectiveness of mental healthcare within

the primary care environment. A systematic review of 36 different interventions concluded that organisational interventions (eg, integration or collaboration) improved the management of chronic mental health conditions, while more simple interventions (eg, practitioner education) were not effective.[6]

One major proposed organisational intervention is the colocation or integration of mental health professionals within the primary care environment to deliver psychological therapies. The colocation of mental health professionals within primarily practices refers to the location of these professionals within the same offices/clinical space as the primary care practice staff. These professionals are full members of the primary healthcare team receiving both self-referrals from patients but also referrals from all other team members, which can include general practitioners (GPs), clinical pharmacists, practice nurses and healthcare assistants.[7] These professionals would generally also be expected to attend meetings within their assigned practice/clinics, and liaise with clinicians across other mental health, social care and physical health.[7] This intervention has achieved increasing levels of traction due to the growing evidence base of the clinical[8 9] and cost effectiveness[10–12] of psychological therapy and the increased patient satisfaction levels of this therapy compared with medication prescription.[13] A recent focus of UK healthcare policy has been to try and implement this intervention within the National Health Service (NHS) with the Government's Five-Year Forward View aiming to integrate 3000 mental health therapists within primary care.[14 15] According to the most recent figures, outlined within Health Education England's workforce strategy (December 2017), there are over 2130 more mental health therapists (83.7%) employed in England.[16] Around 800 of these therapists were subsequently colocated into general practice.[16] However, evidence for this policy is unclear: there are currently no systematic reviews specifically addressing whether the integration of mental health workers (MHWs) into primary care is clinically or cost effective.

Previous systematic reviews have demonstrated improvement in clinical outcomes such as depression through MHWs providing talking therapies (counselling and cognitive–behavioural therapy) in primary care,[9 17 18] although evidence for sustained improvements in social function and mental health is lacking.[19] There is a further lack of general consensus to suggest that cost effectiveness and patient satisfaction levels are improved by this model.[9 18] One review illustrated consultation rates and medication prescription rates decreased and referral rates increased significantly in the short term, but not in the long term.[20] This systematic review therefore aims to assess and appraise the evidence for the clinical effectiveness and cost effectiveness of individual MHWs colocated within primary care practices. This review also focuses on the individual worker effects caused by these mental health professionals rather than those created through collaborative care or consultation–liaison models. It is important to note that this review, through evaluating a broad range of MHWs and patient populations, is expected to find a large degree of heterogeneity between the results of the papers identified. This is as a result of these studies having patients with varied symptomatic severity levels of, and different prevalence of, psychological disorders. Furthermore, different MHWs may be more effective at treating certain disorders or reducing symptom prevalence/severity which could further lead to heterogenous results.

## METHODS

This systematic review was carried out in accordance with the recommendations contained in the Cochrane Handbook for Systematic Reviews of Interventions[21] and reported in line with the Preferred Reporting Items for Systematic Reviews and Meta-Analyses guidelines[22] (online supplemental file 1). The search was conducted in July 2019.

### Eligibility criteria

The PICOS (population; intervention; comparison; outcome; study design) framework was followed in identifying the eligibility criteria. All quantitative studies published before 1 July 2019 were eligible for the review, including randomised controlled trials (RCTs), practice-based evidence studies and cost-effectiveness studies. We included studies with participants of any age and gender. Qualitative studies, studies with full text unavailable, and studies not in English were excluded. Systematic reviews were also excluded but were used for result comparisons. Studies did not need to report a certain outcome measure in order to be eligible for inclusion (ie, General Health Questionnaire (GHQ), Beck's Depression Inventory (BDI) or Clinical Outcomes in Routine Evaluation – Outcome Measure (CORE-OM)). It was also preferable to have studies which included standard GP care as a comparator, but as studies with this were limited in number it was not used to restrict the eligibility criteria.

### Search strategy

The Medline, Embase, PsycINFO, HMIC and Global Health databases were searched via Ovid. The reference lists of related literature reviews were further searched for relevant sources. The search strategy was created by using Medical Subject Headings and search terms related to four elements: (1) mental health disorders, (2) the primary care setting, (3) MHWs and (4) colocation and integration. The Medline search can be found in online supplemental file 2; the full search strategy, for the other databases, is available from the authors on request.

### Study selection

We used Covidence software[23] to select the final set of papers for the systematic review and remove duplicates. Title and abstract, and full-text screening were conducted by two independent reviewers (J-BW and BH), with conflicts being resolved through meetings with a third researcher (GG).

**Table 1** Characteristics of included studies

| Author/date | Country of study | Study design | Sample size | Age of participants | Primary care setting | Mental health worker |
|---|---|---|---|---|---|---|
| Kates et al[27] | Canada | Observational descriptive study | 3550 patients | No age limitation/range not reported | General practice | Counsellor |
| Cigrang et al[26] | USA | Observational descriptive study | 234 patients | 18–87 years | Primary care clinics in a large military medical facility | Doctoral-level clinical psychologists and psychiatrist |
| Boot et al[28] | UK | Randomised control trial | 192 patients | 16 years or above | General practice | Counsellors |
| Abidi et al[29] | Netherlands | Observational descriptive study | ~15 000 patients | No age limitation/range not reported | General practice | Practice mental health nurses and primary care psychologists |
| Evans et al[30] | UK | Observational descriptive study | 65 000 patients | No age limitation/range not reported | General practice | Mental health link worker. |
| Spurgeon et al[31] | UK | Controlled before–after study | 271 patients | No age limitation/range not reported | General practice | Counsellor |
| Pryde and Jachuck[32] | UK | Observational descriptive study | 97 patients | 14–74 years | General practice | Clinical psychologists |
| Magnée et al[34] | Netherlands | Observational descriptive study | 624 477 patients in 2010 1 392 187 patients in 2014. | No age limitation/range not reported | General practice | Mental health nurses |
| McMahon et al[35] | UK | Randomised controlled trial | 62 patients | 18–65 years | Primary care practices | Graduate mental health workers |
| Lester et al[36] | UK | Cluster randomised controlled trial | 368 patients | 18–65 years | General practice | Mental health workers |
| Marks[37] | UK | Randomised controlled trial | 92 patients | No age limitation/range not reported | General practice | Psychiatric nurse therapists |
| Bridges et al[38] | USA | Observational descriptive study | 793 patients | 1–75 years | Primary care clinics | Behavioural health consultant |
| Friedli et al[39] | UK | Randomised controlled trial | 136 patients | No age limitation/range not reported | General practice | Counsellor |
| Milne and Souter[40] | UK | Observational descriptive study | 30 patients | No age limitation/range not reported | General practice | Clinical psychologist |
| Magnée et al[33] | Netherlands | Observational descriptive study | 197 512 patients | 10–65 years | General practice | Mental health nurse |

## Data extraction

Data were extracted using a standardised table (table 1) which listed parameters including type of study design, type of primary care setting, type of MHW, age of participants, country of study implementation, sample size and the main outcomes of the study.

## Risk-of-bias assessment

The Cochrane bias tool was used to assess the risk of bias of randomised trials while the ROBINS-I tool was used to assess the risk of bias of non-randomised trials.[24 25] Studies were not excluded based on risk of bias.

## Data analysis

Pooled analysis within this review proved unfeasible, due to the relatively small number of studies identified and the heterogeneity of effects and outcomes investigated, for example, due to different models of integration or colocation.

## Patient or public involvement

It was not appropriate to involve patients or the public in the design, or conduct, or reporting, or dissemination plans of our research

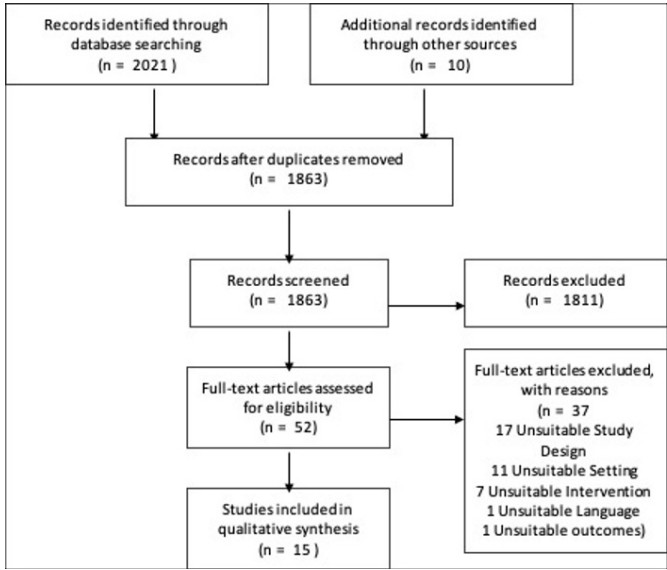

**Figure 1** Preferred Reporting Items for Systematic Reviews and Meta-Analyses flow diagram illustrating the selection process.[22]

## RESULTS
### Study selection
Following removal of duplicates, 1863 titles and abstracts were screened to assess their suitability for inclusion within this review. Subsequently, 52 full-text papers were screened, of which 37 were deemed to be irrelevant. Fifteen papers were identified to be included[26–40] (figure 1).

### Study characteristics
Study characteristics are summarised within table 1. Studies were predominantly conducted in the UK,[28 30–32 35–37 39 40] with others being conducted in the Netherlands,[29 33 34] the USA[26 38] and Canada.[27] The majority of the papers fall either into the category of RCT (five)[28 35–37 39] or observation descriptive studies (nine).[26 27 29 30 32–34 38 40] MHWs involved in the included studies were counsellors,[27 28 31 39] mental health nurse specialists,[29 33 34 37] clinical psychologists and[26 32 40] behavioural health consultants.[38] Various forms of integration model were reported, ranging from a more replacement-structured integration model, which the majority of studies were based on, to a much more collaborative model,[26 38] with one paper assessing both the collaborative and replacement models of care.[29]

### Quality assessment
We used two separate tools to carry out risk-of-bias assessment on all included studies.[24 25] Most were of fair quality, with only a few indicators suggesting a high risk of bias (online supplemental files 3, 4). Five papers were of fair quality,[26 31 33 35 40] three of poor quality,[28 37 39] two of unclear quality,[30 32] five of good quality[27 29 34 36 38] (online supplemental files 5, 6). However, risk of bias was generally higher within RCTs, which had a high risk of performance bias. This is due to the nature of the intervention as participants and personnel in all the RCTs were not

able to be blinded as to the treatment they received. For non-randomised papers, missing data incurred the highest degree of bias. Quality assessment for included studies is available from the authors on request.

### Outcomes
The study outcomes are summarised in online supplemental file 7.

#### Mental health outcomes
There were three main scales found among the papers included, GHQ,[27 28] BDI,[35 39] CORE-OM.[31 36] The other scales reported are Outcome Questionnaire 45 (OQ-45),[26] Hospital Anxiety and Depression Scale (HADS),[31] A Collaborative Outcomes Resource Network (ACORN),[38] Computerised Revised ClinicalInterview Schedule (CRCIS),[39] Short Form 36 Health Survey (SF-36),[31] Center for Epidemiologic Studies Depression Scale (CESD),[27] Hamilton Depression Rating Scale (HDRS17),[35] Brief Symptom Inventory (BSI),[39] Montgomery-Asberg Depression Rating Scale (MADRS)[35] and Skidmore anxiety stigma scale (SASS),[35] although some papers included more than one scale and others did not report a scale at all. Eight papers reported a clinical benefit to integrating MHWs into primary care practices[26–28 31 32 37 38 40] compared with the usual care that a GP would provide, while four papers did not report on these outcomes.[29 30 33 34] For example, one study reported that symptoms of individuals in the integrated MHW programme improved substantially, with average changes of 17.6 in CESD Score and 5.7 in GHQ Scores.[27] Moreover, prior to the study commencing, individuals were assessed as reaching the threshold for inclusion based on CESD or GHQ Scores. After treatment the total number of individuals exceeding threshold values had decreased significantly, 73% (CESD) and 82% (GHQ) ($p < 0.005$). A paper by Cigrang et al reported similar outcomes. OQ-45 Scores decreased significantly for patients with more than one appointment. There was also a significant reduction in OQ-45 Scores between patients that had either had 1, 2 or 4 or more appointments. This was also maintained during follow-up (p range 0.032–0.001). However, three papers suggested that clinical improvements are not likely to be significantly greater than with usual GP care.[35 36 39] One of these papers provides evidence that no significant differences, compared with standard care, for any of the mental health outcomes measures at either the 3 or 9 month periods were able to be seen.[39]

#### Cost outcomes, psychotropic drug usage and service utilisation
In terms of service utilisation and referrals, there are seven papers that used these factors as one of their main evaluation outcomes.[26–28 30 31 34 38] Only two papers provided comprehensive results data in terms of referrals,[27 30] and these reported different findings. One[30] reported an increase in referrals to mental health specialists among minority communities and the other[27] reported a decrease in overall patient referrals. Service

utilisation rates varied between studies, with two reporting no difference in GP appointment rates between intervention and control groups.[28 34] Conversely, another study showed a mean increase in GP appointments (6.8–8.4)[26] and another, investigating chronic conditions, showed a significantly reduced rate of health services utilisation (GP appointments, home visits and patient referrals to mental health specialists) after 12 months.[31] Two papers investigated the association between the intervention and the levels of psychotropic drug usage,[28 33] and a further two papers evaluated the general costs associated with the intervention.[39 40] Psychotropic drug usage was found to remain stable during and after the implementation of the intervention in one study,[33] while another reported significantly fewer prescriptions as a result of the intervention.[28] Finally, there were only two studies identified from the search that providing data evaluating the cost effectiveness of the intervention.[39 40] Both of these studies found increases in overall costs. One study reported that the MHW intervention cost an additional £162 per patient per year (indirectly and directly) compared with usual GP care for the first 3 months, although in the preceding 6 months costs were £87 less per patient.[39] The other study reported overall increases in drug costs over the period; however the results varied between groups within the study. The costs of psychotropic and other drugs reduced in the improved coper group (11 individuals), whereas they increased significantly in the remaining 11 individuals which accounted for the overall group trend. Reductions in costs for the improved copers were due less drug usage, less hospital referrals and less GP visits.

### Patient satisfaction of the intervention

Five interventions were evaluated in terms of patient satisfaction levels.[27 28 35 36 38] These papers illustrate a high level of satisfaction for this MHW integration and three of these papers further specify that this intervention was associated with significantly higher levels of patient satisfaction when compared with normal GP care.[27 28 36] For example, Kates et al[27] illustrate that through using the Consumer SatisfactionQuestionnaire (CSQ) Scale consumers had an overall satisfaction of 92%, while Lester et al[36] using the same scale report that intervention practices patients had higher mean levels of satisfaction than those in control practices (p=0.023). The remaining two papers showed no significant differences in satisfaction levels between intervention and non-intervention patients.[35 38]

### DISCUSSION
### Summary

There is evidence from this review that the integration of MHWs in primary care practices provides meaningful mental health benefits to varied populations, including minority groups and those with comorbid chronic diseases.[26–28 31 32 37 38 40] However, there is insufficient evidence to suggest that these improvements are clinically

significant when compared with usual care. Similarly, the evidence base surrounding the cost effectiveness of this intervention is mixed, with no common consensus as to whether integration of MHWs is more cost effective than standard GP care. Service utilisation and drug prescription rates vary considerably between studies, while referral rate changes suggest reductions in the overall burden on specialist care and improved access for minority groups. Patient satisfaction levels were consistently high in studies that measured this construct.

### Comparison with existing literature
### Mental health outcomes

The delivery of mental health treatments within the primary care environment has been consistently shown to be clinically effective,[18 19 27 41] although this is most distinct within the short term (1–6 months). Primary care policy supporting integration of MHWs in primary care practices suggests an expectation that the integration of MHW's is likely to be correlated with decreased severity and a reduction of symptoms in patients with mental health problems. However, studies which included general care as a comparator provide no overall consensus as to whether there is any significant clinical benefit of integration over standard GP care. This finding may be as a result of standard GP care already having elements of GP-delivered counselling. This would make significant changes in symptoms less likely, as the treatment being delivered by the MHW would not be completely different.[41 42] Furthermore, the non-significant change in symptom reductions in certain studies may be attributed to the fact that the patients in these studies have less serious mental health problems. Consequently, it would be unlikely for there to be a large effect size when integrating an MHW, as symptom levels and their severity are already low or moderate and so the scope for potential improvement is limited. For instance, in the paper by McMahon et al,[35] the baseline characteristics of the patients for numerous mental health outcome scales were usually rated as moderate, BDI 26.2 (moderate depression), HDRS17 18.1 (20 or higher is classed as moderate severity), MADRS 24.3 (moderate depression). Moreover, it is possible that the small effect size of integrating MHWs within primary care practices was as a consequence of the included primary care practices being self-selecting. Therefore, the practices included would have had an established interest in mental health service provision, and as a consequence were more likely to be participants in the included studies, while also being expected to already perform well in terms of general mental healthcare.

### Cost outcomes, psychotropic drug usage and service utilisation

Quality improvement strategies have generally been focused primarily on reducing the burden of disease within the healthcare system where they are implemented.[43–45]

However, cost containment has become necessary in recent years with healthcare systems worldwide being under strain from rising healthcare costs, recession and

the burden of an ageing population.[46–48] Integration of mental health practitioners has been devised as a solution to this problem as it is hypothesised that this could lead to reductions in resource utilisation and in health service utilisation, including fewer referrals to mental health specialists, fewer GP consultations and reduced drug prescriptions. However, this study and others assessing the overall resource utilisation of the intervention suggest that it is in fact associated with increased cost.[9 18 39 40] There is a study that suggests that the intervention does not lead to increased costs, but this had a small sample size.[9]

One area where there is agreement between studies, concordant with the findings of this review, is in referral changes associated with MHW integration. Decreases in the frequency of referrals to mental health specialists are well demonstrated in similar studies focused on both specific MHWs[9 18] and MHWs in general.[20 49] If integrated MHWs successfully reduce GP appointment rates, this can allow GPs more time to treat other patients.[39] The results of one study found some evidence that referral to an MHW led to a small reduction in GP consultations, although with a small effect size.[20] This contrasts with the findings of our review, which finds no clear agreement between studies on whether integration of MHWs increased or decreased GP consultations.

### Patient satisfaction

In recent years, the growing emphasis on the importance of patient and practitioner satisfaction has been frequently highlighted in terms of healthcare suitability and delivery.[50] Although satisfaction may not be a determinant indicator of success of an intervention,[51] satisfaction levels can ultimately decide whether utilisation rates of the service are maintained and whether practitioners continue using a service over usual care. In accordance with the results of this review, patient satisfaction for MHW intervention in primary care has been shown to be consistently significantly higher than that of standard GP care.[18–20 49]

### Limitations

We did not include non-English language studies; therefore, relevant evidence from non-English studies may have been missed. This review has been unable to use a substantial number of papers related to the effectiveness, cost effectiveness and other potential benefits of delivering specific psychological and psychosocial treatments within primary care. This is as a result of them not being focused specifically on the effectiveness of the MHW itself. It was also difficult to include studies relating to collaborative care as it was not possible to determine whether the study's effects were owed to the MHW; some studies had more than one MHW within this collaborative care process which further increased uncertainty. Other studies were focused on introducing certain interventions within primary care practices which were irrelevant to the review, including those of phone treatment, internet

treatment and self-help. With more time and resource availability, this study would have also included a broader literature search that removed the database search terms relating specifically to integration, collocation and mental illnesses. This is hypothesised to be able to find slightly more relevant studies to review but will also increase the number of papers having to be screened exponentially. The addition of mixed method studies and qualitative studies could have given a broader perspective and additional findings; however, this paper was solely interested in quantitative evaluations.

### Implication for further research

Further research is needed to compare the clinical effectiveness of the integrated MHW with standard GP care with only five papers in this study comparing this. There also needs to be a greater emphasis on reporting what the usual care provided by primary care practitioners is, in order to accurately assess the clinical differences in care provided. Rigorous cost-effectiveness studies need to be created, which should assess cost effectiveness to both the service provider and the broader healthcare system, and illustrate any societal costs associated with the intervention.

### Implication for practice and policy

The WHO has advocated in recent years that the primary healthcare environment is the optimal environment for the treatment of a plethora of conditions, including mental health disorders.[3] Thus, a multitude of structural interventions and changes have been proposed to fulfil this policy change effectively. One of the changes that has been proposed and that this review has focused on is that of the integration of an onsite MHW within primary care practices. This review has found that the integration of an MHW is correlated in some studies with decreased severity and a reduction of symptoms and therefore leads to better healthcare outcomes for patients with a range of mental health problems. This supports existing evidence that providing mental healthcare within the primary care setting is beneficial and that increasing resource expenditure on MHWs will lead to beneficial impacts to the mental health of the populations exposed to them. However, there is insufficient evidence from this review to suggest that MHWs within primary care practices significantly elevate the quality of mental healthcare, compared with standard GP care. Likewise, there is little evidence to suggest that the intervention is more cost effective, although this may be as a consequence of there being few studies examining this. Larger scale introductions of this intervention, and further RCTs, will enable more substantial and rigorous evaluations of clinical and cost effectiveness. Until then, a degree of caution is needed in investing significant resources to implement this intervention. Furthermore, other possible policy and quality improvement recommendations relating to management of mental health problems should be investigated. For instance, a review has suggested that mental health

treatments within secondary care may be more effective than those delivered in primary care, thus providing more resources to this environment may be more beneficial.[52]

## CONCLUSIONS

The NHS in England plans to increase the number of MHWs within primary care. This study suggests that further research is needed to evaluate both the clinical effectiveness and cost effectiveness of this policy to ensure it is a good use of health system resources. Clarity is also needed on the best type of health professional to take on MHW roles within primary care.

**Contributors** J-BW was involved with collecting the data, analysing the data, interpreting the data and drafting the manuscript. GG contributed to the design, data collection and revised the all drafts of the manuscript. BH conceived the idea for the study, contributed to the design and data collection, and revised all drafts of the manuscript. AM was involved in revising the manuscript. All authors read and approved the final manuscript.

**Funding** This article presents independent research commissioned by the National Institute for Health Research (NIHR) under the Applied Health Research (ARC) programme for North West London. The views expressed in this publication are those of the author(s) and not necessarily those of the NHS, the NIHR or the Department of Health.

**Competing interests** BH and AM are general practitioners working in the NHS. We have not been prompted or paid by anyone to write this article.

**Patient consent for publication** Not required.

**Provenance and peer review** Not commissioned; externally peer reviewed.

**Data availability statement** Data are available in a public, open access repository. Data are available upon reasonable request. All papers and data in this review can be found using the submitted search strategy. All papers can be found through OVID. The search strategies for the listed databases are available upon reasonable request, as is the quality assessment data.

**ORCID iDs**
Jean-Baptiste Woods http://orcid.org/0000-0002-7210-0719
Geva Greenfield http://orcid.org/0000-0001-9779-2486
Azeem Majeed http://orcid.org/0000-0002-2357-9858
Benedict Hayhoe http://orcid.org/0000-0002-2645-6191

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
