## [Reviewer comments · BMJ Open]

ARTICLE DETAILS

TITLE (PROVISIONAL)	The clinical effectiveness and cost-effectiveness of individual mental health workers co-located within primary care practices. A systematic literature review.
AUTHORS	Woods, Jean-Baptiste; Greenfield, Geva; Majeed, Azeem; Hayhoe, Benedict

VERSION 1 – REVIEW

REVIEWER	Sebastian Hinde Centre for Health Economics University of York United Kingdom
REVIEW RETURNED	16-Jul-2020

GENERAL COMMENTS	Thank you for inviting me to review this paper. Overall I found it to be a reasonable and well conducted review but it could be presented more clearly. I have reviewed this as a health economist. My comments are as follows: - The title needs some reference to it being a review- Given the international breadth of the included studies some overview of the nature of primary care in the different countries seems necessary- It would be good to have a clearer statement regarding the intervention, what it is, who it is for, and what it is an alternative to.- The systematic review referenced in the first paragraph of the introduction (p5 line 19) is from 2003 which seems quite old in this context, are you happy it is still relevant- Does 'chronic conditions' p5 line 22 refer to chronic mental health conditions or chronic conditions more generally?- The last 2 sentences of the second paragraph of the introduction (p5 lines 34 to 43) are very unclear, specifically regarding the Five Year Forward view. The reference used (13) isn't to the Five Year Forward view, and even if it was it has been replaced by the NHS Long Term Plan as the Forward View was 2014 and the Long Term Plan 2019. Was there any reference to this intervention in the Long Term Plan? It would also be helpful to know if the Five Year Forward View referenced any studies in its policy recommendation, and whether the hope of 3000 therapists was achieved.- Could you make it clearer when the review was conducted, was it from the 1st July 2019?- Given the wide range of quality of studies you report at the top of page 11 it isn't clear why you don't stratify the results by the quality of the studies? It currently feels like you flag the variation and then give all of the studies equal weighting when presenting the results.- Only one reference is made to the cost of the mental health therapists (page 12 line 40), is this because this is the only study that reported a cost?
--

	- In addition to being in the title a lot of reference is made to the cost-effectiveness of the intervention. However, there doesn't appear to be any reference to it in any of the results section. Is this because no studies specifically mentioned it? Given you present it as a major part of the review it is a little unclear of whether no studies considered it or the ones that did were of poor quality. You state that 'no common consensus as to whether integration of MHWs is more cost-effective than standard GP care' but that isn't clear from the results. - The first sentence of the discussion summary needs references - You state 'It is generally accepted that the integration of MHWs is likely to be correlated with...' but there is no reference given for this. - Is there any case to be made that the effect of the intervention was small because the GP practices included were self-selecting and therefore of better quality than the average GP practice in their provision for mental health services? - You state that 'the non-significant change in symptom reductions in certain studies may be attributed to the fact that the patients in these studies have less serious mental health problems', is there any way you can back this up from the papers? How severe are the patient populations?
--	--

REVIEWER	Helena Tuomainen Warwick Medical School, University of Warwick, UK
REVIEW RETURNED	07-Aug-2020

GENERAL COMMENTS	The paper is a systematic review of the literature to establish the clinical effectiveness and cost-effectiveness of individual mental health workers co-located within primary care practices. Considering that NHS in England is planning to increase the number of MHWs within primary care, this paper is of importance. The paper is very well written and I have only a few significant comments. Title: add term systematic review I would be interested to know the age range of included patients. Is this information available in the papers? If yes, I would add this information to Table 1. Were the findings at all dependent on age of patient? In the methods section, suggest adding reference to the main mental health outcome measures, and whether included studies had to include specific outcome measures or not. Suggest reporting (main) outcome measures used in studies at the beginning of section Outcomes/mental health outcomes; i.e. moving this information from the end of the section to the beginning. P. 13, line 34-35: It would be helpful if you clarify what you mean with "within the short term". Minor comments/edits: P. 4 Line 8: is focussed on (not if) p. 5 Collocated or co-located? Use one or the other consistently p. 6 Line 12: add commas: with varied symptomatic severity levels of, and different prevalence of, psychological disorders Line 41: Remove 'however' p. 8, line 11: remove 'of these'
--

	Table 1: Column sample size: use term 'patients' or 'practices' consistently in every cell. Also in Study design: use 'Observational descriptive study' for last two studies in table. p. 11 line 15/16: nature of the intervention Lines 28- 31/32: Improve the sentence as it gives the impression that the total number of studies is 12, although it is 15. p. 12 line 10/11: In terms of (add s) p. 12, line 14: reported different (gap) p. 13 Line 24-25: Improve last sentence in paragraph. Suggest: Patient satisfaction levels were consistently high in studies that measured this construct. p. 13, Lines 41-47. This sentence is very long and could benefit from cutting into two sentences, after the term counselling. p. 14 Line 20: that it is in fact associated (present tense) Line 21: There is a study that suggest that the intervention does not lead to increased costs (present tense) Line 26-31: Again, suggest cutting sentence in half, to improve readability. Supplement 4: Clinical or clinical outcomes – use terminology consistently
--	---

VERSION 1 – AUTHOR RESPONSE

Reviewer: 1

1. The title needs some reference to it being a review-

The title has been reformatted to include this term. 'The clinical effectiveness and cost-effectiveness of individual mental health workers co-located within primary care practices. A systematic literature review.'

2. Given the international breadth of the included studies some overview of the nature of primary care in the different countries seems necessary-

Whilst we accept that the precise definition and structure primary care differs between countries considered in included studies, a broader definition of primary care, as a first point of contact healthcare system providing care for individuals with both mental and physical health problems, applies in all of these settings. Consequently, we did not feel that detailed analysis of the primary care systems in existence in the various countries included was necessary in this report.

3. It would be good to have a clearer statement regarding the intervention, what it is, who it is for, and what it is an alternative to.

A description for the intervention has now been included 'The co-location of mental health professionals within primary practices refers to the location of these professionals within the same offices/ clinical space as the primary care practice staff. These professionals are full members of the primary healthcare team receiving both self-referrals from patients but also referrals from all other team members, which can include GPs, clinical pharmacists, practice nurses and health care assistants (7). These professionals would generally also be expected

to attend meetings within their assigned practice/ clinics, and liaise with clinicians across other mental health, social care and physical health (7).'

- 4. The systematic review referenced in the first paragraph of the introduction (p5 line 19) is from 2003 which seems quite old in this context, are you happy it is still relevant**

We believe that the evidence within this is still relevant and there are no recent reviews which cover the different types of mental health intervention as well as this one does.

- 5. Does 'chronic conditions' p5 line 22 refer to chronic mental health conditions or chronic conditions more generally?-**

We have amended this so it states 'chronic mental health conditions'.

- 6. The last 2 sentences of the second paragraph of the introduction (p5 lines 34 to 43) are very unclear, specifically regarding the Five Year Forward view. The reference used (13) isn't to the Five Year Forward view, and even if it was it has been replaced by the NHS Long Term Plan as the Forward View was 2014 and the Long Term Plan 2019. Was there any reference to this intervention in the Long Term Plan? It would also be helpful to know if the Five Year Forward View referenced any studies in its policy recommendation, and whether the hope of 3000 therapists was achieved.-**

The reference has been amended to show correct evidence. Unfortunately, there were no studies referenced within the five year forward view's policy recommendation, only a couple case studies for which no quantitative information is provided. Evidence has been added regarding how many therapists have been added to the workforce in England, although these figures are from December 2017 as no further data appears to be available.

- 7. Could you make it clearer when the review was conducted, was it from the 1st July 2019?-**

Date has been added at the start of the methods section.

- 8. Given the wide range of quality of studies you report at the top of page 11 it isn't clear why you don't stratify the results by the quality of the studies? It currently feels like you flag the variation and then give all of the studies equal weighting when presenting the results.**

As we note in the manuscript (quality assessment p11) the nature of the intervention meant that included RCTs were inevitably subject to bias (it was not possible to blind individuals as to the treatment received). Whilst there is clearly some variability in quality of included studies, we did not feel this was sufficient to affect significantly the findings in terms of the thematic analysis we carried out.

- 9. Only one reference is made to the cost of the mental health therapists (page 12 line 40), is this because this is the only study that reported a cost?-**

Indeed, no other studies at the time of the search were found to have any reliable cost data related specifically to mental health therapists.

10. In addition to being in the title a lot of reference is made to the cost-effectiveness of the intervention. However, there doesn't appear to be any reference to it in any of the results section. Is this because no studies specifically mentioned it? Given you present it as a major part of the review it is a little unclear of whether no studies considered it or the ones that did were of poor quality. You state that 'no common consensus as to whether integration of MHWs is more cost-effective than standard GP care' but that isn't clear from the results.

The results section has been expanded to clearly mention that there were only 2 papers found that explicitly investigate the cost-outcomes of the intervention and also to provide a more detailed summary of what these papers identified. This amended section is as follows: 'Finally, there were only two studies identified from the search that providing data evaluating the cost-effectiveness of the intervention (39,40). Both of these studies found increases in overall costs (40). One study reported that the MHW intervention cost an additional £162 per patient per year (indirectly and directly) compared with usual GP care for the first three months, although in the preceding 6 months costs were £87 less per patient (39). The other study reported overall increases in drug costs over the period, however the results varied between groups within the study (40). The costs of psychotropic and other drugs reduced in the improved copers group (11 individuals), whereas they increased significantly in the remaining 11 individuals which accounted for the overall group trend (40). Reductions in costs for the improved copers were due less drug usage, less hospital referrals and less GP visits (40).

11. The first sentence of the discussion summary needs references

References have now been included within this sentence.

12. You state 'It is generally accepted that the integration of MHWs is likely to be correlated with...' but there is no reference given for this.-

The sentence has been altered in order to be clearer to the reader. We have amended this to being 'Primary care policy supporting integration of MHWs in primary care practices suggests an expectation that the integration of MHW's is likely to be correlated with decreased severity and a reduction of symptoms in patients with mental health problems.'

13. Is there any case to be made that the effect of the intervention was small because the GP practices included were self-selecting and therefore of better quality than the average GP practice in their provision for mental health services?-

We have decided to accept this suggestion. In the manuscript we have written: 'Moreover, it is possible that the small effect size of integrating MHWs within primary care practices was as a consequence of the included primary care practices being self-selecting. Therefore, the practices included would have had an established interest in mental health service provision, and as a consequence were more likely to be participants in the included studies, whilst also being expected to already perform well in terms of general mental health care.'

- 14. You state that ‘the non-significant change in symptom reductions in certain studies may be attributed to the fact that the patients in these studies have less serious mental health problems’, is there any way you can back this up from the papers? How severe are the patient populations?-**

Evidence has now been provided in the manuscript. ‘For instance, in the paper by McMahon et al., (2007) the baseline characteristics of the patients for numerous mental health outcome scales were usually rated as moderate, BDI 26.2 (moderate depression), HDRS17 18.1 (20 or higher is classed as moderate severity), MADRS 24.3 (moderate depression).’

Reviewer: 2

- 1. I would be interested to know the age range of included patients. Is this information available in the papers? If yes, I would add this information to Table 1. Were the findings at all dependent on age of patient?**

Age has now been included into the table as this was available within some of the studies. However, the age ranges of the studies are all similarly broad and we did not feel that age was likely to have been a confounding factor.

- 2. In the methods section, suggest adding reference to the main mental health outcome measures, and whether included studies had to include specific outcome measures or not.**

We have amended this to make reference to the fact that specific outcome measures were not needed for inclusion.

- 3. Suggest reporting (main) outcome measures used in studies at the beginning of section Outcomes/mental health outcomes; i.e. moving this information from the end of the section to the beginning.**

This has now been moved as suggested.

- 4. P. 13, line 34-35: It would be helpful if you clarify what you mean with “within the short term”.**

This has now been clarified to be between 1 to 6 months.

- 5. Minor comments/edits: -**

- P. 4 Line 8: is focussed on (not if)
- p. 5 Collocated or co-located? Use one or the other consistently
- p. 6 Line 12: add commas: with varied symptomatic severity levels of, and different prevalence of, psychological disorders
- Line 41: Remove ‘however’
- p. 8, line 11: remove ‘of these’
- Table 1: Column sample size: use term ‘patients’ or ‘practices’ consistently in every cell. Also in Study design: use ‘Observational descriptive study’ for last two studies in table.
- p. 11 line 15/16: nature of the intervention

- Lines 28- 31/32: Improve the sentence as it gives the impression that the total number of studies is 12, although it is 15.
- p. 12 line 10/11: In terms of (add s)
- p. 12, line 14: reported different (gap)
- p. 13 Line 24-25: Improve last sentence in paragraph. Suggest: Patient satisfaction levels were consistently high in studies that measured this construct.
- p. 13, Lines 41-47. This sentence is very long and could benefit from cutting into two sentences, after the term counselling.
- p. 14 Line 20: that it is in fact associated (present tense)
- Line 21: There is a study that suggest that the intervention does not lead to increased costs (present tense)
- Line 26-31: Again, suggest cutting sentence in half, to improve readability.
- Supplement 4: Clinical or clinical outcomes – use terminology consistently

Thank you - all suggestions implemented.

VERSION 2 – REVIEW

REVIEWER	Sebastian Hinde Centre for Health Economics, University of York, UK
REVIEW RETURNED	30-Sep-2020

GENERAL COMMENTS	The authors have returned a very thorough and well considered set of revisions. I have no further comments on the paper and am therefore happy to recommend it for acceptance.
--

REVIEWER	Helena Tuomainen University of Warwick, United Kingdom
REVIEW RETURNED	19-Oct-2020

GENERAL COMMENTS	I am happy with the revisions, and have no more comments or feedback.
---